# Health system and quality of care factors contributing to maternal deaths in East Java, Indonesia

**Mohammad Afzal Mahmood**[1,2]\*, **Hendy Hendarto**[3,4], **Muhammad Ardian Cahya Laksana**[3,4], **Hanifa Erlin Damayanti**[3,4], **Mohammad Hud Suhargono**[5], **Rizki Pranadyan**[3,4], **Kohar Hari Santoso**[6], **Kartika Sri Redjeki**[7], **Baksono Winard**[3,4], **Budi Prasetyo**[3,4], **Jorien Vercruyssen**[8], **John Robert Moss**[1], **Peng Bi**[1], **Syarifah Masitah**[9], **Warsiti**[4], **Aldilia Wyasti Pratama**[10], **Erni Rosita Dewi**[10], **Charity Hartika Listiyani**[10], **Ismi Mufidah**[11]

1 School of Public Health, University of Adelaide, Adelaide, SA, Australia, 2 Faculty of Medicine, Universitas Airlangga, Surabaya, Indonesia, 3 Faculty of Medicine, Department of Obstetrics and Gynaecology, Universitas Airlangga, Soetomo Teaching Hospital, Surabaya, Indonesia, 4 Soetomo Teaching Hospital, Surabaya, East Java, Indonesia, 5 Bangil General Hospital, Pasuruan District, Jawa Timur, Indonesia, 6 Department of Health, Jawa Timur, Surabaya, Indonesia, 7 Department of Health, Surabaya, Indonesia, 8 Obstetrics Department, Antwerp University Hospital, Edegem, Belgium, 9 Wahab Syaharani Hospital, Samarinda, Indonesia, 10 Faculty of Medicine, School of Midwifery, Universitas Airlangga, Surabaya, Indonesia, 11 Department of Health, Kutai Kartanegara, Kalimantan Timur, Indonesia

\* afzal.mahmood@adelaide.edu.au

**Data Availability Statement:** The data cannot be uploaded as additional files and cannot be sent to a public repository considering ethical restriction as

## Abstract

Despite most Indonesian women now receiving antenatal care on the nationally recommended four occasions and being delivered by skilled birth attendants, the nation's maternal mortality ratio (MMR) is estimated as 177 per 100,000 live births. Recent research in a rural district of Indonesia has indicated that poor service quality due to organizational and personnel factors is now a major determinant of this high MMR. The present research is an in-depth analysis of possible health service organizational and quality of care related causes of death among 30 women admitted to a peak referral hospital in a major Indonesian city. Despite their condition being complex or deteriorating, most of these women arrived at the hospital in a state where it was feasible to prevent death with good quality care. Poor application of protocols, poor information flow from frontline hospitals to the peak referral hospital, delays in emergency care, and delays in management of deteriorating patients were the main contributing factors to these deaths. Pyramidal referrals also contributed, as many women were initially referred to hospitals where their condition could not be effectively managed. While generic quality improvement measures, particularly training and monitoring for rigorous application of clinical protocols (including forward planning for deteriorating patients) will help improve the situation, the districts and hospitals need to develop capacity to assess their local situation. Unless local organisational factors, staff knowledge and skill, blood and blood product availability, and local reasons for delays in providing care are identified, it may not be possible to effectively reduce the adverse pregnancy outcomes.

per University of Adelaide Human Research Ethics Committee (HREC, The University of Adelaide). This is for the reason that the data contains Sensitive Patient Information. Request for the data could be sent to HREC (email: hrec@adelaide.edu.au) Level 4, Rundle Mall Plaza, 50 Rundle Mall, Adelaide, SA 5005, Australia, with attn to Yvette Wijnandts, Research Ethics Officer. HREC reference for this research is H-2018-217.

**Funding:** The authors received no specific funding for this work. For travel to Indonesia, the Australian researcher used general research-tied funding from the University of Adelaide. Local expenses for research-related meetings and local travel were covered by general funds provided by Airlangga University.

**Competing interests:** The authors have declared that no competing interests exist.

## Introduction

In Indonesia, the national maternal mortality rate (MMR) remains persistently high, being estimated as 177 per 100,000 live births [1] despite 87% pregnant women across the country now receiving antenatal care (ANC) for the nationally recommended four times and 84% now delivering by a skilled birth attendant [2].

In the province of East Java in 2017, 97% and 78% of women had at least one and four ANC visits (respectively) to skilled attendants [2]. 95% of women had a skilled attendant at delivery [3]. Despite this access, many women suffered from complications; and the reported MMR in East Java was 91 per 100,000 live births in 2017, with 534 maternal deaths. Over the last few years, the East Java MMR has stagnated with little reduction [3]. Yet internationally there is a steadily decreasing MMR everywhere except in sub Saharan Africa [4]. However, this overall decline has been slow, likely due to delays in access to care and to poor quality of care—and for many around the globe, this care is still 'too little too late' and for some 'too much too soon' [5].

For a further reduction in MMR and for improved pregnancy outcomes, it is important that the root-causes of this stagnation become known. Maternal death audits are needed to assess health services factors that contribute to poor outcomes [6]. Research undertaken in the Kutai Kartanegara district of Indonesia has demonstrated that the root-causes lie mainly within system factors leading to poor quality of care [7]. Such information about the health system organisation and its association with quality of care is needed, as improvement in quality of care is essential to improving maternal and newborn outcomes [8].

The research in Kutai Kartanegara highlighted the causes of poor quality as being: poor risk management in primary care, fractured communication across levels of care, delayed referrals, delayed or inappropriate referrals and transfers, inadequate forward planning, poor monitoring of complications, poor interpersonal communication, and ineffective use of medical records [7]. A large majority of maternal deaths in Indonesia now occur within hospitals. It became clear more than a decade ago that effective intrapartum care is a priority for improving pregnancy outcomes [9]. The persistent high maternal mortality and the current high number of maternal deaths in hospital require that the underlying reasons are investigated in depth.

The research reported here has addressed the question 'what are the organisational and quality of care root-causes of the persistently high maternal mortality?' by analysis of maternal deaths at a tertiary care hospital.

## Method

A systems approach and root-cause analysis were used to critically analyse quality of care, with a focus on understanding system failures resulting in poor quality of care [10]. Donabedian's [11] quality of care framework was used to define interventions for quality improvement; and Berwick's [12] theory of continuous improvement provided a framework for change.

Root-cause analysis of 30 maternal deaths was conducted at one of the two peak referral hospitals in East Java province. Cases were selected from amongst the 70 maternal deaths that occurred at this peak referral hospital between 1 January and 31 December 2017. Maternal deaths of all 15 women who were residents of the city where this hospital is based were reviewed. Another 15 deaths of women from selected districts across East Java were also reviewed; these included 4 from a district where the MMR is relatively higher and where a second phase of intervention-research is planned with the aim of decreasing the MMR; and the remaining 11 deaths in this group were from districts strategically selected for distance, causes of death and MMR. 30 deaths were reviewed in light of our research in East Kalimantan

province of Indonesia informing that a detailed review of these many deaths allows an in-depth understanding of the locally relevant root causes to maternal deaths.

The data for the root-cause analysis included medical records at the referral hospital, any available information provided to the referral hospital by the frontline hospitals from where these women were transferred, and any information compiled by the district and sent to the provincial department of health.

At least two researchers, drawn from a medically trained doctor with health system review expertise, an obstetrician, and a midwife, reviewed each case. They wrote detailed summaries, including notes on the terminal illness. The analysis focused on organizational and management factors, team environment, individual staff knowledge, and skills and practices. This data included information on whether care was integrated or fragmented, and on communication, timeliness, referral, facilitation of transfers to hospitals and its follow-up, and organizational factors such as availability of senior consultants. Each case summary was then reviewed by other team members and discussed at review meetings. The analysis also focussed on identifying significant risks and how these were managed, and whether they were managed promptly or if there was delay or other failure. For each woman, a meeting was then held to discuss the findings of the individual reviewers and, if possible, to develop a consensus about the identified root-causes. Four members, consisting of an Indonesian obstetrician, an Indonesian district health maternal health coordinator, a midwife and a health system specialist, discussed whether the knowledge and/or skill of any organisation or health care provider played a role. A factor was considered as contributing to death if all members who reviewed the case agreed. This process required revisiting the medical notes in detail.

In 2017, a total of 70 women died at the peak referral hospital. To understand the overall context, a more basic audit of all deaths was conducted, in addition to the root-cause analysis of 30 of these 70 deaths. This basic audit consisted of a review of information on demography, risk factors, information provided by the hospital from where the patient was transferred (if available), and cause of death.

The research was approved by the University of Adelaide Human Research Ethics Committee (H-2018-217) & by the Referral Hospital (1294/KEPK/VII/2019) where the research was conducted.

## Setting

The location for the in-depth analysis of 30 deaths is a 1,444-bedded referal hospital. It has a well-established midwifery unit with 45 obstetricians. In 2017, there were 1,370 births, out of which 793 were by C-Section. The hospital has outpatient, inpatient, laboratory, operating theatre and emergency obstetric care services, radiology (including an ultrasound facility), and a blood bank on the premises. The MMR of the city in which the peak referral hospital is located is 73/100,000 live births, compared to the provincial MMR of 91. Generally, women cannot book themselves for delivery at this peak hospital, except in an emergency when this is the nearest hospital to their home. As it is a referral hospital, almost all women arrive on referral or as a transfer from a frontline hospital in the province. Hence, women admitted to this hospital are already at high risk.

The city where this hospital is based had a total population of 2,874,699 in 2017, spread across 31 subdistricts and 154 villages. In that year, there were 42,822 births and 34 maternal deaths in this city. There were 59 hospitals in total, including 55 with obstetric units; and 63 primary health care centres (Puskesmas); and 183 obstetricians and 285 private midwifery clinics. The hospital under study is one of two peak referral hospitals in the province.

Administratively, East Java (Jatim) Province consists of 29 districts and 9 cities, with 8,501 villages across the districts. As of 2017, there were 964 government primary care centres, out of which 623 provide basic inpatient care while 341 provide outpatient care only. There were 380 hospitals in the province: 116 government and 264 private. In 2017, total live births in Jatim were 570,819. The total population of the province was 39,292,972 [13].

In East Java, there were 529 maternal deaths in 2017 and 522 in 2018. In 2018, live births were 570,819. The MMR differed substantially across the districts from a high of 301 to less than 50 in six districts. 89% of these maternal deaths occurred at a hospital (20% at private and 69% at government hospitals). 54.4% of these deaths occurred during the postnatal period and 20% during delivery [14]. The main causes of maternal death in the province were pre-eclampsia/eclampsia (30.1%), haemorrhage 24.7%), pre-existing cardiovascular disease (10.8%) and infections (4.8%). Among the remaining, concomitant illnesses contributed substantially to death. The districts with an MMR higher than 100 tended to cluster in the South East of the province and in the West. These clusters may be a reflection of the distance from tertiary care in the peak referral hospitals, and of sociodemographic factors.

## Results

69 of the 70 women who died at this peak referral hospital had been first seen at a lower level [frontline] hospital; 36 had spent at least 24 hours at a frontline hospital before being transferred on. Of the 70 deaths that occurred at this hospital, 55 were to women whose place of residence was outside of the city where it is located. The causes of death for these 70 women included PE/Eclampsia (27%), haemorrhage (17%), infection (30%) and others (30%—including CVD, ARDS, embolism, SLE, TB, cancers, SLE etc.).

While 5 out of the 70 who died had arrived directly at the peak hospital, 65 were transferred from lower level frontline hospitals. A majority, 41 out of 65, who were first admitted to a frontline hospital were transferred on the same day they arrived. However, 21 out of 65 were at the frontline hospital for many days before they were transferred to the peak hospital. For example, one woman was transferred after 16 days at the frontline hospital and died 23 hours after arrival. Two other women were at a frontline hospital for 6 and 5 days, and died 12 hours and 9 hours (respectively) after transfer.

18 of these 70 women were delivered by Caesarean Section at a frontline hospital and then transferred to the peak hospital. 12 women suffered bleeding at or before arriving at the frontline hospital. However, information about any blood transfusion prior to arriving at the peak hospital was available for only 2 of these 12.

### Root-cause analysis of 30 maternal deaths at the peak referral hospital

The root-cause analysis was conducted on 30 deaths which occurred at the peak referral hospital in 2017 out of the total of 70 deaths in that location. This sample included all 15 deaths from the city (also designated as a district) in which that hospital is based, while the other 15 were selected from those transferred from other districts in the province. Eighteen of these 30 women were less than 30 years of age, 10 were 30–35 years and 2 were above 35 years of age. 22 of these 30 women were gravida 2 or 3; 7 were gravida 1; while only 1 was gravida 4.

20 women out of the 30 had a concomitant illness, of whom 8 had CVD (including left to right shunts, Eisenmenger syndrome, heart failure), 4 had pre-existing hypertension, 2 had hyperthyroidism, and one had TB, one HIV/AIDS, one diabetes, one renal disease, one retinoblastoma, and one obesity. While these concomitant illnesses contributed to the complexity and severity, not all concomitant illnesses were the main cause of death. They varied from mild to very severe disease.

Overall, 15 out of 30 of these women required an emergency C-Section (Table 1); 9 at the peak referral hospital and 6 at the frontline hospitals. As almost all the pregnant women at this hospital are transferred from other hospitals, they tend to be suffering from obstetric complications and many are in a critical condition. Overall, of the 1,370 women who were delivered at this hospital in 2017, 793 were delivered by C-Section. Out of the 15 women who came from another districts, two thirds (10/15) required an emergency C-Section. Out of those 15 who arrived from the district where referral hospital is located only 4 required an emergency C-Section.

Organisational and personnel factors contributed to all 30 deaths. Table 2 summarises the findings of the in-depth review of the maternal death audit, data collected by the district departments of health, and the clinical information contained in the medical records at the referral hospital.

The organisational, staff and personal/family factors presented in Table 2 are explained below:

**Poor organisation/management (e.g. schedules, rosters, links with blood bank).** In 6 out of 30 deaths, poor organisation or management was considered to be a factor. The frontline hospitals from where these patients presented appeared to lack a strategic management plan to effectively treat patients who arrived with complications and/or deterioration due to poor quality ANC. Similarly, the frontline hospitals appeared to have no planning for post-discharge follow-up. The women attended primary care, risk factor(s) were identified, but follow-ups were not made. For example, for an 18 year old who had a recent history of miscarriage, the primary care provider did not plan and conduct any follow-up.

**Lack of policy/protocol/guidelines.** Protocols for follow-ups were not available or not followed in the primary care and hospital settings. In the case of at least 5 women, it appeared that the protocols for managing deterioration were either not available or not used effectively. For example, a woman kept receiving the same ineffective medicine for many days. For another woman, it appeared that the frontline hospital did not have an agreed protocol for managing pulmonary oedema and pulmonary embolism. In another case, the protocols for inverted uterus, heavy bleeding and severe anaemia and hypovolemia were either not available or not applied.

**Inadequate staff and inadequate access to senior clinical staff.** Overall, inadequate staff numbers were not a major concern. For only 3 out of 30 women did inadequate staff numbers appear to contribute to delays and/or inadequate management. Discussions with the research team and with the referral hospital staff indicated that in many cases basic protocols are available and that the staff are experienced. However, many junior doctors working at the frontline hospitals are probably not adequately trained. Nevertheless, only in one case did it appear that the staff at the bedside might have benefitted from advice from a senior staff member.

**Delay in emergency response.** Emergency response delays at the frontline and peak hospitals contributed to at least 5 out of 30 deaths. For example, it took two hours at a frontline hospital before a woman, who arrived in a critical condition, was seen.

**Table 1. Emergency C-Section intervention.**

| C–Section | From within the city-district where referral hospital is based (n = 15) | From other districts & cities of East Java (n = 15) |
|---|---|---|
| Not performed | 10 | 4 |
| Planned | 0 | 1 |
| Emergency | 5 | 10 |

**Table 2. Health services organisational, staff and personal factors (How the women were managed by the system–not only by the peak hospital).**

| ORGANISATIONAL FACTORS (more than one factor may be applicable to each maternal death) | City where hospital is based (n = 15) | Other cities & districts of East Java (n = 15) | Total (n = 30) |
|---|---|---|---|
| Poor Organisation/Management | 3 | 3 | 6 |
| Lack of Policy/Protocol/Guidelines | 1 | 4 | 5 |
| Insufficient Numbers of Staff | 3 | 2 | 3 |
| Inadequate Access to Senior Clinical Staff | 0 | 1 | 1 |
| Delay in Emergency Response | 2 | 3 | 5 |
| Delay in Procedures | 0 | 6 | 6 |
| Poor Process for Sharing Information | 0 | 3 | 3 |
| Delay in Access to Test Results | 0 | 1 | 1 |
| Pyramidal Referral | 8 | 7 | 15 |
| **STAFF** | | | |
| Knowledge and Skills Lacking-ANC Staff | 11 | 6 | 17 |
| Knowledge and Skills Lacking-Hospital Staff | 7 | 9 | 16 |
| Delay in Emergency Response | 3 | 2 | 5 |
| Poor Communication | 4 | 5 | 9 |
| Failure to Seek Supervision/Help | 0 | 0 | 0 |
| Failure to Follow Best Practice | 4 | 9 | 13 |
| Lack of Recognition of Seriousness | 5 | 3 | 8 |
| **ENVIRONMENT** | | | |
| Geography (i.e. distance caused delays or difficulties with provision of care) | 0 | 0 | 0 |

**Delays in procedures** at the frontline and referral hospitals contributed to 7 out of 30 deaths. For example, there were crucial delays in a hysterectomy in managing blood loss; and also in a termination of pregnancy where it appeared that the staff waited for the baby to mature to 34 weeks. In one case, there was a delay in initiating a laparotomy because no ventilator was available.

**Poor process of sharing information with the patients/families.** In only three cases did inadequate information sharing with the family members contribute to ineffective management. For example, family members of one woman appeared to be not well informed by the hospital staff about the need for continued hospital inpatient care.

**Pyramidal referral.** The government health care referral policy states that, if a referral/ transfer is required, a primary care provider or centre should refer the patient to the next level health facility (a frontline hospital, level B or level C hospital) and not directly to a higher level (Level A) hospital. This pyramidal referral system appeared to contribute to poor outcomes for 15 out of 30 deaths (Table 2). Some women had been referred initially to hospitals where their condition could not be effectively managed. These women might have achieved better outcomes if they had been referred directly to the peak referral hospital. A woman was transferred from primary care to a frontline hospital, but the frontline hospital was not well prepared to care for her in a thyroid crisis. Another woman was kept at the frontline hospital for 15 days although the staff there could not find the source of infection. Another woman should have been shifted to the referral hospital on the first or second day at the frontline hospital, as she suffered from cardiac arrest twice and her renal function deteriorated.

**Knowledge and skills of antenatal care staff.** For 17 women, poor ANC was a contributory factor. It appeared that the risk management was not optimum, which requires effective communication, appropriate referrals and timely transfers, and follow-ups. Even hypertension, which is quite common in Indonesian obstetrics, was inadequately managed. The records of a

woman, whose BP was high and who had had a history of hypertension for several years, indicated that, during the pregnancy, a thorough assessment with a full medical work up to identify the cause of the hypertension was not conducted. She had not been on antihypertensive medication even though her BP started rising early in the pregnancy. There were other cases where risks were not assessed adequately. For example, a woman who had TB signs and symptoms was not assessed for that disease.

**Inadequate application of knowledge and skills & failure to follow best practice at hospitals.**   In 16 out of 30 cases, inadequate knowledge and skills were among the contributory factors; and for 13 out of these 16 there was failure to follow evidence-based best practice. For example, a woman suffered from post C-Section cardiac arrest despite being otherwise healthy, thereby raising questions about the quality of anaesthesia and pre-operative management. For another, the care quality was questionable as information about management of a cardiac left to right shunt was inadequately reported. Another woman was kept at the frontline hospital for fourteen days post C-Section, suffered from severe infection leading to sepsis, and her hypertension could not be managed adequately. Another woman received 4 ANC consults at a hospital, but developed complications of HTN/PE as her BP was not well managed. It seemed that, if not earlier, she did have high blood pressure at the last ANC, without any specific management, which was ten days before her death. During her terminal illness, she had pulmonary oedema on arriving at the frontline hospital.

**Poor communication.**   This contributed to 9 out of 30 deaths. Poor communication is a major concern and appeared to affect all levels of care For many, it appeared that the ANC provider did not adequately inform the woman or her family about the risks; and during the labour there was no communication, via phone or other media, between the primary care staff attending the labour and the hospital for advice about management. Communication by hospital specialists to the women to help them understand the risks and their need for treatment at a hospital was inadequate in at least 7 cases. For example, in one case, inadequate communication between ANC and hospital, within the hospital (obstetrics unit, pulmonologist, internist), and between the frontline hospital and the referral hospital contributed to ineffective management.

**Lack of recognition of seriousness.**   This contributed to 8 deaths. For example, there was a lack of recognition of the seriousness of a case of uncontrolled hypertension, with no plan or urgency about the situation. At the hospital, her BP fluctuated and she was discharged when her infection was, most probably, not well managed. Another woman died after being on anti-hypertensive medication prescribed by an obstetrician for the preceding three months. There was no follow up to assess how well the hypertension was controlled with that medication. When she arrived at the district hospital, a C-Section was performed due to her being post-date and having oligohydramnios.

## Discussion

A large majority of Indonesian women are now delivered by skilled birth attendants, having received ANC the nationally recommended number of times. Front-line and peak referral hospitals are adequately resourced in the context of the state of national economic development and are available as a back-up for higher-risk situations. While much has been achieved in the provision of safe obstetric care at each of these levels, the MMR indicates that there is still substantial room for improvement. Access to skilled attendants for ANC and delivery does not necessarily lead to a good health outcome unless the actual health care is of optimal quality [7,15]. The scope for improvement lies especially in the organisation and management of the health care system.

The basic audit and the root-cause analysis of deaths at the referral hospital complemented each other, providing a deeper understanding of the overall situation and of the woman's risk profile. Despite a poorly performing primary care system, a majority of the women arrived at either the frontline or the peak referral hospital in a condition where death might have been prevented with better quality hospital care. As noted in our study in the Kutai Kartanegara district of East Kalimantan [7], a better performing health care system with measures to address organisational and personnel factors is the key to addressing this persistently high maternal mortality in Indonesia.

This research has identified various aspects on the spectrum from ANC via Intrapartum Care to Post Natal Care (PNC) that require improvement. In addition to the poor application of knowledge at both primary care and in hospitals, a major concern is the challenge in providing coordinated care between ANC in primary care, acute care in frontline hospitals and emergency care at the referral hospital for women suffering from complications and/or deterioration. At present, their care is organisationally disjointed.

The protocol for audits and analysis, developed to cover safe motherhood services, timeliness of care [16], quality of care concepts [12] and organisational, staff and environmental factors [7,17], provided a framework with which to assess the contributory factors that need addressing for improved pregnancy outcomes and better maternal health. An important limitation was we did not have access to the medical records at the frontline hospitals. Nevertheless, the in-depth analysis by experienced researchers and clinicians of information provided by the frontline hospital to accompany the women being transferred, detailed medical records from the peak referral hospital and death audit notes provided a wealth of information about system inadequacies and what needs to be done.

The city where the peak referral hospital is based had an MMR of 73 compared to the provincial MMR of 91, with 23 of 38 districts having much higher MMR than this city [3]. Varying death rates across the districts need to be further investigated through in-depth analysis in those locations. The current research indicates that many frontline hospitals provided care to women who would have benefitted from early transfer to a referral hospital. Despite their deteriorating condition and despite an inadequate skill level at these frontline hospitals, the women were not transferred or were transferred late. While geography is probably a factor for those who resided at a greater distance, the main reasons for delayed or absent transfer were probably a lack of understanding by the frontline hospital staff of the limitations and weaknesses of their hospital, and a lack of service links and professional interaction between frontline hospitals and the referral hospital. The delays in transferring are mainly due to organisational factors and knowledge and skill deficits and not due to patient concerns about affordability, as under the universal health insurance cover a woman would not incur a financial penalty for a transfer. The districts need to carefully assess their local situation. Unless local factors, which might be organisational, staff knowledge and skill, blood and blood product availability, or relatively longer delays in providing care, are identified and acted upon, it may not be possible to address the poorer pregnancy outcomes through generic measures such as an increased number of antenatal care visits or encouraging women to deliver at a hospital.

The role of concomitant illnesses in leading to poor pregnancy outcomes and maternal death is becoming more important. As maternal mortality decreases in developing countries, the relative role of concomitant illnesses in poor pregnancy outcomes is increasing [18]. At the province level, a large proportion of deaths are categorised as 'others' in reports by the districts (East Java Department of Health 2018). There is a need to describe 'others' in detail and to report exactly what conditions (such as TB, Dengue, Malaria, HIV, CVD) these women suffered from. Unless these causes are clearly defined, it will be challenging to retrain staff, equip

facilities appropriately, and provide the protocol-based integrated care necessary to manage concomitant disease during pregnancy.

This research found less than optimal care for effective management of hypertension and pre-eclampsia/eclampsia. The role of ineffective management, ineffective communication, inappropriate or poorly facilitated referrals and lack of follow-up has also been found in other research [19]. This in-depth review indicates that the primary care component of the system, particularly the government community health centres and private midwives, appears weak in terms of identifying and managing risks. Key roles of the government primary care centres are to provide outreach care and care close to the homes of pregnant women, to identify those who are at risk, and to facilitate care through treatment, referral and follow-ups. This situation requires local research and in-depth audits to identify locally prevalent organisational, social and personal risk factors.

Ineffective management of inter-hospital transfers and delayed access to blood transfusion have been reported as factors contributing to deaths of women suffering from postpartum haemorrhage in low resource settings [20]. This requires review of the situation in each district, particularly where haemorrhage is causing more deaths. A major concern is the inadequate information provided by the frontline hospitals to the referral hospital. Effective care at the referral hospitals and better outcomes require detailed and better information flow between hospitals. Women are arriving at the referral hospital without the necessary detailed information about their clinical condition and why they have been referred/transferred.

30% women who died at this peak referral hospital were reported to have sepsis. This appears to be a reflection of the overall inadequate quality of care, as the cause of sepsis and poor pregnancy outcomes associated with sepsis lie in poor preventive measures and inadequate management of peri-partum infections [21]. Hospitals need to assess their infection rates and analyse whether infection control measures are in place and effective. Additionally, the district health authorities and the health centres in the area need to be informed by the hospital that the woman has been treated for an infection and that follow-ups are required to make certain that the infection is fully managed.

The organizational (hospital, primary care services) and the personnel factors played key roles in determining quality of care, and ultimately contributed to a large number of deaths. Inadequate skills and poor application of protocols, inadequate communication, and delays in provision of emergency care and/or delays in decisions about management of deteriorating patients were the main contributing factors to sub-optimal quality. The Indonesian health system is progressing well towards universal access to care and an increasing number of women now deliver at health facilities [22]. Poor quality of care despite adequate physical access is a cause of poor pregnancy outcomes [23]. This concern about health care quality and its impact is being raised by others as well [5,24].

This in-depth review of data on 30 women who died at the referral hospital found that access to health services, availability of staff for ANC and availability of senior staff at the hospitals were adequate. It has been noted in other research that access to obstetricians and doctors to attend women arriving with postpartum haemorrhage and severe pre-eclampsia is a major concern [25]. However, in the present research almost all of the deaths occurred despite access to sufficient human resources at the tertiary hospital. In contrast, inappropriate or delayed referral, delayed transfers, a lack of coordination amongst levels of care, and poor forward planning for managing any potential deterioration were issues that affected many outcomes.

An inappropriate interpretation of the pyramidal referral policy, whereby primary care providers referred the deteriorating patient to a frontline hospital when she actually needed referral hospital care, was a contributing factor in many of the deaths. A review by Singh et al [26]

pointed out that "[some of the] referrals are haphazard and a pregnant woman at high risk or with complications did not get the required EmOC [Emergency Obstetrics Care] and had to go through several referrals before reaching the appropriate institution". Poorly judged pyramidal referrals, disjointedness, and lack of understanding about the precipice of disaster (a stage of deterioration/complication which is beyond the capability of a particular facility and when the patient must be transferred or she may die) cascaded the problem. Lower levels in the referral chain must recognise their limits in managing a patient who is in trouble. To save her life, they must transfer her to facilities better able to cope. In many cases, frontline hospitals appeared not to have the skills or resources to manage potential complications that may arise after a C-Section. Quite a few women developed complications which could not be managed at the frontline hospital. There was no information in the notes about whether the frontline hospital consulted with the referral hospital on how to manage once the complication arose. Delayed transfers from the frontline hospital were a major concern. Many women died within only a few hours of arriving at the referral hospital; and often they had been at a frontline hospital for many days. The frontline hospitals appeared to be working beyond the capabilities of their staff and resources. Primary care providers and frontline hospitals need to understand the concept of 'knowing their limitations'. Many women remained at a frontline hospital when their condition had already deteriorated beyond the degree to which the staff were able to manage. Monitoring of health services for their readiness to provide effective services for women who arrive with complications has been emphasised in recommendations to improve quality of care [27,28].

The factors undermining quality are present in primary care as well. The inadequate quality of care and inconsistent approach to managing risks such as pre-eclampsia in primary care have been noted previously as a major factor in maternal mortality [18]. These factors undermining quality of primary care were obvious in this study as well. There was evidence of poor communication between primary care and hospital; and almost no feedback from the hospitals to primary care settings. Another reason for poor quality in primary care is probably the lack of skills in assessing risks. Management of hypertension is particularly a major area of concern.

The main recommendations arising from this research in East Java are summarized in Table 3.

Table 3. Recommendations to address the factors contributing to maternal mortality.

| CONTRIBUTING FACTORS | RECOMMENDATIONS |
|---|---|
| Failure to follow protocols and best practices | District Health Departments should institute a system of closer supervision & support, and conduct in-depth audits to understand the weaknesses in the local system that result in poor risk management and delays in referrals and transfers to hospitals. ANC quality requires much attention, and the staff need support and supervision for protocol-based care and sustained management of risks. The frontline hospitals' midwifery units need protocols for evidence-based care to be displayed throughout. Grand rounds need to be instituted. The hospital should be engaged in reviewing deaths, near misses and deterioration. In-service training needs to be tailored to the local factors, such as prevention and management of sepsis, and organisational factors affecting provision of quality care |
| Pyramidal Referrals | This policy needs to be discussed in depth so that it is better understood, with a better understanding by the staff that patients can be transferred to higher level hospitals (by-passing the frontline hospital) provided their clinical condition is carefully described and noted. Primary care staff must be trained to know about the capacity and capabilities of the hospitals in their area, and should have access to locally relevant protocols about where to and how to refer/transfer patients. |
| Disjointed Care | Protocols should be developed for early assessment and subsequent categorisation into high, intermediate and low risk, with each category having a clear plan about where to deliver. Staff should be trained in the use of these protocols, including information about the capability of each of the district hospitals in terms of what services are available. |
| Ineffective Communication | Focussed training should be conducted for midwifery supervisors and heads of primary care centres to communicate and develop delivery plans with the woman, document the condition in sufficient detail, and use the relevant hotline, with calls to the hospital before and during the transfer. |

## Conclusion

Organisational and personnel factors, at both primary care and in the hospitals, now contribute to almost all maternal deaths in Indonesia. While ANC in primary care is still of less than optimum quality, most deaths are preventable if the hospital care is of optimum quality. There is a need to develop capacity to assess the local situations across the hospitals and districts in terms of organisational and personnel factors that undermine quality of care. Effective use of available infrastructure and human resources, improvement in organisational management practices, improved communication and organisational links between primary care, and front-line and referral hospitals, and support and supervision for prompt protocol-based care could potentially lead to substantial improvement in pregnancy outcomes and decrease maternal mortality. There is also a need to plan and provide timely and effective care for concomitant illnesses.

## Acknowledgments

The research team members express their sincere thanks to the management and staff of the participating referral hospital, and to the districts and provincial departments of health in East Java, Indonesia. Our sincere thanks are also extended to the Women's and Children's Hospital, Adelaide and the Flinders Medical Centre, South Australia, which both provided placement opportunities for the principal investigator and the Indonesian obstetrician members of the research team to enable them to learn about quality of care factors within the obstetric departments of these hospitals. The research team is thankful to research assistant Fidya Panorama Damayanti for her support in organising team meetings and interaction with relevant.

## Author Contributions

**Conceptualization:** Mohammad Afzal Mahmood, Hendy Hendarto, Muhammad Ardian Cahya Laksana, Hanifa Erlin Damayanti, Rizki Pranadyan, Kohar Hari Santoso, Kartika Sri Redjeki, Baksono Winard, Budi Prasetyo, Jorien Vercruyssen, Peng Bi, Syarifah Masitah, Ismi Mufidah.

**Data curation:** Muhammad Ardian Cahya Laksana, Hanifa Erlin Damayanti, Warsiti, Aldilia Wyasti Pratama, Erni Rosita Dewi, Charity Hartika Listiyani.

**Formal analysis:** Mohammad Afzal Mahmood, Hendy Hendarto, Muhammad Ardian Cahya Laksana, Hanifa Erlin Damayanti, Mohammad Hud Suhargono, Rizki Pranadyan, Jorien Vercruyssen, John Robert Moss, Aldilia Wyasti Pratama, Erni Rosita Dewi, Charity Hartika Listiyani, Ismi Mufidah.

**Methodology:** Mohammad Afzal Mahmood, Muhammad Ardian Cahya Laksana, Hanifa Erlin Damayanti, Mohammad Hud Suhargono, Kohar Hari Santoso, Kartika Sri Redjeki, Baksono Winard, Budi Prasetyo, Jorien Vercruyssen, John Robert Moss, Peng Bi, Syarifah Masitah, Warsiti, Ismi Mufidah.

**Project administration:** Mohammad Afzal Mahmood, Muhammad Ardian Cahya Laksana.

**Resources:** Rizki Pranadyan, Kohar Hari Santoso, Kartika Sri Redjeki.

**Supervision:** Mohammad Afzal Mahmood, Hendy Hendarto, Muhammad Ardian Cahya Laksana.

**Writing – original draft:** Mohammad Afzal Mahmood, Muhammad Ardian Cahya Laksana.

**Writing – review & editing:** Mohammad Afzal Mahmood, Hendy Hendarto, Muhammad Ardian Cahya Laksana, Hanifa Erlin Damayanti, Mohammad Hud Suhargono, Rizki Pranadyan, Kohar Hari Santoso, Kartika Sri Redjeki, Baksono Winard, Budi Prasetyo, Jorien Vercruyssen, John Robert Moss, Peng Bi, Syarifah Masitah,  Warsiti, Aldilia Wyasti Pratama, Erni Rosita Dewi, Charity Hartika Listiyani, Ismi Mufidah.

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
