## [Decision Letter · Decision Letter 0]

13 Jul 2020

PONE-D-20-09737

Health System and Quality of Care Factors Contributing to 522 Maternal Deaths in East Java, Indonesia

PLOS ONE

Dear Dr. Mahmood,

Thank you for submitting your manuscript to PLOS ONE. After careful consideration, we feel that it has merit but does not fully meet PLOS ONE’s publication criteria as it currently stands. Therefore, we invite you to submit a revised version of the manuscript that addresses the points raised during the review process.

I agree with the comments made by Reviewer #2, that overall the paper needs to be shorter, more focused on the most important data, and edited for syntax, grammar and spelling. I look forward to seeing the revised version where I may provide further comments.

We look forward to receiving your revised manuscript.

Kind regards,

Emily A Hurley, M.P.H., Ph.D.

Academic Editor

PLOS ONE

Journal Requirements:

2. In your Methods section, please provide additional information about the methodology used .

Please ensure you have provided sufficient details to replicate the analyses such as:

a) a description of any inclusion/exclusion criteria that were applied to cases inclusion in your analysis ,and a justification of the number of cases included;

b) more information on how the data shown in Table 5 were obtained (how experts opinion were collated, and how consensus was achieved).

4. Thank you for stating the following above the Acknowledgments Section of your manuscript:

'FUNDING

The research was conducted using the resources of the University of Airlangga and the Department of Health,

with no outside funding.'

'The authors received no specific funding for this work.'

'The authors have declared that no competing interests exist.' 

We note that one or more of the authors are employed by a commercial company: Consultant Obstetrician

6. Your ethics statement must appear in the Methods section of your manuscript. If your ethics statement is written in any section besides the Methods, please move it to the Methods section and delete it from any other section. Please also ensure that your ethics statement is included in your manuscript, as the ethics section of your online submission will not be published alongside your manuscript.

7. We note that Figure 1 in your submission contains map images which may be copyrighted.

We require you to either (a) present written permission from the copyright holder to publish these figure specifically under the CC BY 4.0 license, or (b) remove the figure from your submission:

b. If you are unable to obtain permission from the original copyright holder to publish these figure under the CC BY 4.0 license or if the copyright holder’s requirements are incompatible with the CC BY 4.0 license, please either i) remove the figure or ii) supply a replacement figure that complies with the CC BY 4.0 license. Please check copyright information on all replacement figures and update the figure caption with source information. If applicable, please specify in the figure caption text when a figure is similar but not identical to the original image and is therefore for illustrative purposes only.

8. Please ensure that you refer to Figure 2 in your text as, if accepted, production will need this reference to link the reader to the figure.

Reviewers' comments:

Reviewer's Responses to Questions

**Comments to the Author**

1. Is the manuscript technically sound, and do the data support the conclusions?

Reviewer #1: Yes

Reviewer #2: Partly

2. Has the statistical analysis been performed appropriately and rigorously? 

Reviewer #1: Yes

Reviewer #2: N/A

3. Have the authors made all data underlying the findings in their manuscript fully available?

Reviewer #1: Yes

Reviewer #2: Yes

4. Is the manuscript presented in an intelligible fashion and written in standard English?

Reviewer #1: Yes

Reviewer #2: No

5. Review Comments to the Author

Reviewer #1: This paper presents an extraordinary overview of the systemic processes that have resulted in an unacceptable level of maternal mortality in East Java. All too often we are lured into a false sense of security by the ongoing introduction of advancing technologies that serve to improve clinical outcomes, not recognizing that the salutary impact of that process is the proper implementation of those technologies and, perhaps most importantly, the development of a clinical system that accurately evaluates an individual and provides for the proper clinical setting and staff that can best utilize technology and expertise to assure optimal outcomes. This paper clearly describes a situation where, despite the availability of appropriate clinical technology, a broken system serves to maintain an inappropriate rate of maternal mortality. The paper is easy to follow but suffers slightly from occasional poor style and syntax, as well as an occasional spelling error, scattered throughout the manuscript. Nonetheless, this paper presents critically important information to our readers and to obstetrical providers and administrators worldwide.

Reviewer #2: Overall , the paper is very long. I do not think there is substantial additional value from the details of the 522 deaths, or the 70 deaths. The information in these sections has been widely demonstrated in previous published literature.

I suggest removing these sections, and concentrating on telling a much more concise story about the deaths that were analysed in detail. The introduction and the discussion should also be substantially reduced.

Minor points:

The manuscript would benefit from review by a native English speaker to correct minor errors eg in abstract and introduction “now receiving antenatal care for an adequate number of times”

Line 68 “In East Java in 2016, 97% and 78% of women had at least one and four ANC visits (respectively) to skilled attendants”. Please describe the proportion of women who have a skilled attendant at delivery

Line 70 : “Recently, the East Java MMR has stagnated with little reduction.” Please reference this data

Line 72 Please reconsider this statement. The paper by Hogan shows a steadily decreasing MMR everywhere except sub Saharan Africa, specifically demonstrated in Webfigure 6c. Maternal mortality ratio (MMR) per 100, 000 live births, high MMR regions

Line 82 “Better performing health systems and good quality care are needed to reduce these deaths, the majority of which now occur at the time of labour and delivery and in the postpartum period (Renfrew 2014). “ This sentence appears to simply repeat the previous sentence and could be removed.

Line 123 – line 142. I am unclear if this section refers to the current paper, or the previous paper by the same author. If it refers to the previous paper, I do not think it needs to be included with this degree of detail.

Table 1 – this is a very detailed table that doesn’t provide much additional information. An attempt should be made to describe the uncertainty, or variability around these numbers from year to year.

Comparisons regrading rates of death by age or parity, as has been done throughout the manuscript are not very useful without knowing the age/parity of the entire population of women giving birth.

269: concomitant illness. This list varies from very mild disease to very severe disease which was very likely to contribute to maternal death ( left to right shunt and Eisenmengers disease). It

276: this cesarean section rate seems consistent with the very high cesarean rate for the hospital, as described in line 156 , and so does not support the argument that the women were in critical condition.

6. PLOS authors have the option to publish the peer review history of their article (what does this mean?). If published, this will include your full peer review and any attached files.

Reviewer #1: **Yes: **Prof. Lee P. Shulman, Feinberg School of Medicine of Northwestern University, Chicago, Illinois, USA.

Reviewer #2: No

---

## [Author Response · Author response to Decision Letter 0]

29 Aug 2020

The research team wishes to thank the editors and the reviewers for their in-depth reviews and very helpful suggestions. This has helped us to revise the paper, which now focuses on the root-cause analysis, highlighting causes of poor quality of care leading to maternal deaths. The changes are described in detail in the 'response to reviewers' document. The changes with reference to journal's requirement are described in the cover letter.

---

## [Decision Letter · Decision Letter 1]

17 Feb 2021

Health System and Quality of Care Factors Contributing to Maternal Deaths in East Java, Indonesia

PONE-D-20-09737R1

Dear Dr. Mahmood,

We’re pleased to inform you that your manuscript has been judged scientifically suitable for publication and will be formally accepted for publication once it meets all outstanding technical requirements.

Kind regards,

Emily A Hurley, M.P.H., Ph.D.

Academic Editor

PLOS ONE

Additional Editor Comments (optional):

Reviewers' comments:

Reviewer's Responses to Questions

**Comments to the Author**

1. If the authors have adequately addressed your comments raised in a previous round of review and you feel that this manuscript is now acceptable for publication, you may indicate that here to bypass the “Comments to the Author” section, enter your conflict of interest statement in the “Confidential to Editor” section, and submit your "Accept" recommendation.

Reviewer #1: All comments have been addressed

2. Is the manuscript technically sound, and do the data support the conclusions?

Reviewer #1: (No Response)

3. Has the statistical analysis been performed appropriately and rigorously? 

Reviewer #1: (No Response)

4. Have the authors made all data underlying the findings in their manuscript fully available?

Reviewer #1: (No Response)

5. Is the manuscript presented in an intelligible fashion and written in standard English?

Reviewer #1: (No Response)

6. Review Comments to the Author

Reviewer #1: (No Response)

7. PLOS authors have the option to publish the peer review history of their article (what does this mean?). If published, this will include your full peer review and any attached files.

Reviewer #1: No

---

## [Editor Report · Acceptance letter]

19 Feb 2021

PONE-D-20-09737R1 

Health System and Quality of Care Factors Contributing to Maternal Deaths in East Java, Indonesia 

Dear Dr. Mahmood:

I'm pleased to inform you that your manuscript has been deemed suitable for publication in PLOS ONE. Congratulations! Your manuscript is now with our production department. 

Kind regards, 

on behalf of

Dr. Emily A Hurley 

Academic Editor

PLOS ONE